# Diagnosis and Assessment of Dental Caries Using Novel Bioactive Caries Detecting Dye Solution

**DOI:** 10.3390/biomedicines11020500

**Published:** 2023-02-09

**Authors:** Shashirekha Govind, Amit Jena, Sushanta Kumar Kamilla, Neeta Mohanty, Rachappa M. Mallikarjuna, Triveni Nalawade, Sanjay Saraf, Naseer Al Khaldi, Salma Al Jahdhami, Vinay Shivagange

**Affiliations:** 1Department of Conservative Dentistry and Endodontics, Institute of Dental Sciences, Siksha ‘O’ Anusandhan (Deemed to be) University, Bhubaneswar 751003, India; 2Department of Conservative Dentistry and Endodontics, Sriram Chandra Bhanja Dental College & Hospital, Cuttack 753007, India; 3Department of Physics, Institute of Technical Education and Research, Siksha ‘O’ Anusandhan (Deemed to be) University, Bhubaneswar 751003, India; 4Department of Oral Pathology and Microbiology, Institute of Dental Sciences, Siksha ‘O’Anusandhan (Deemed to be) University, Bhubaneswar 751003, India; 5Department of Paediatric Dentistry, Child Dental Health, Oman Dental College, Mina Al Fahal, Muscat, Wattayah 116, Oman; 6Department of Oral Biology and Oral Pathology, Oman Dental College, Mina Al Fahal, Muscat, Wattayah 116, Oman; 7Consultant Prosthodontist, Al-Nahdha Hospital, Al Khuwair, Muscat 133, Oman; 8Specialist, Oral and Maxillofacial Surgeon, Al-Nahdha Hospital, Al Khuwair, Muscat 133, Oman; 9Department of Conservative Dentistry and Endodontics, Saveetha Dental College, Chennai 600077, India

**Keywords:** dental caries, diagnostic imaging, bioactive caries detecting dye solution, 3D scanner, caries excavation, chemomechanical caries removal, dentin hypersensitivity, minimal intervention dentistry, pain perception

## Abstract

Background: The goal of materials should be early caries detection, removal of carious lesions, and reduction of dentin hypersensitivity. Thus, the study aims to determine the efficacy of a bioactive caries detecting dye (BCD) for the diagnosing and mechanical removal of occlusal and proximal dental caries. Methods: Patients with occlusal (A1, A2) and proximal carious lesions (B1, B2) were treated with the rotary technique and BCD solution on 120 teeth (*n* = 60 for each). Group 1: Excavation was performed using diamond points. Group 2: 0.5 mL of BCD solution was scrubbed for 20 sec and excavation was performed with a sharp spoon excavator. Post-excavation cavity volume analysis was performed using a 3D scanner. The time required, VAS for pain, VAS for facial expression, and sound eye motor scoring were scored during excavation. Post-restoration evaluation was performed at 3, 6, and 12 months (FDI criteria). Results: The chi-square test revealed that the A1 (197.90 30.97 s) and B1 (273.06 69.95 s) had significantly less mean procedural time than the A2 (292.13 44.87 s) and B2 (411.86 88.34 s). BCD (A2, B2) group showed good patient acceptance, less pain during caries excavation VAS (*p* = 0.001, FACE (*p* = 0.001), and SEM (*p* < 0.001) analysis than the rotary group. There was a statistically insignificant difference between groups immediately (*p* = 0.235), (*p* = 0.475) and after 24 h (*p* = 0.561), (*p* = 0.688). Color score, hardness of excavated surface, and caries removal score for occlusal and proximal groups showed insignificant differences between the groups. BCD group showed significantly less mean caries excavated volume for the occlusal group (*p* = 0.003) as compared to the proximal group (*p* = 0.417) evaluated by 3D scanner. Evaluation of restoration after 3-, 6-, and 12 months intervals (Occlusal caries group (*p* = 0.247), (*p* = 0.330), and (0.489) and Proximal caries group (*p* = 0.299), (*p* = 0.594), and (0.494)) was acceptable for both the groups. Conclusion: BCD helps in identification of dental caries clinically, radiographically, and in effective removal of denatured teeth with less pain or sensitivity.

## 1. Introduction

Determining and managing early dental caries involves extensive techniques. Caries will be evident clinically when discoloration occurs in the enamel and dentin. However, initial caries is visible as a white opaque lesion on the enamel surface after drying [1]. A carious lesion consists of a surface zone (active caries) and a subsurface porosity zone (mineral loss) that extends from the tooth’s outer surface to the inner circum-pulpal dentine; a zone of degradation, a zone of microbial invasion (infected dentine), followed by a zone of demineralization, a zone of dentine sclerosis, and fatty degeneration (affected caries and transparent dentin). Non-cavitated active lesions have a higher risk of progression and pulpal involvement than inactive non-cavitated lesions [2]. However, biomimetic agents such as Tri- Calcium Phosphate, CPP-ACP, fluorides, and synthetic nano-hydroxyapatite can cause remineralization in non-cavitated teeth in early stages [3].

Modern restorative dentistry provides minimal intervention alternatives to traditional rotary tissue removal methods (handpieces, steel, carbide, and polycarbonate burs). Mechanical excavation (hand excavators, airabrasion, ultrasonic, and sono-abrasion), chemomechanical methods (Caridex, Carisolv, enzymes, and Carie-Care), and chemochemical techniques involve gentle excavation of chemically softened carious dentin [4]. Despite the fact that chemomechanical caries removal (CMCR) products require a significantly longer working time and have a lower specificity, clinical studies have shown that they are reliable [5]. However, chemochemical solutions can be justified to overcome the drawbacks of CMCR products. According to our previous studies result, the BCD SEM analysis showed 77.66 percent dentinal tubule occlusion and effective caries removal. Cytotoxicity tests revealed BCD is nontoxic and biocompatible. Antimicrobial tests (zone of inhibition) showed BCD > chitosan > chlorhexidine > Carie-Care > Xenetix350 > nHA. Stereomicroscopy showed BCD mechanically removed caries without dye in the dentinal tubules, unlike Carie-Care. BCD had 80–85% dentinal tubule occlusion (SEM analysis) and Carie-Care 0% [6]. As a result, this research focuses on the novel BCD as a chemochemical solution and investigates: (a) The efficient removal of caries and the amount of time taken to excavate after application of the BCD solution with the conventional rotary technique with diamond points; (b) using 3D scanner, the volume of caries excavated for each group; (c) evaluation of pain perception by VAS, VAS for facial expression, and sound eye motor scale (SEM) scoring recorded during excavation of caries and 24 h post-restoration; (d) evaluation of restoration after 3, 6, and 12 months. All parameters were carried out on the occlusal and proximal caries. The null hypothesis stated that there is no difference in working time and patient acceptance between the conventional rotary group and BCD group.

## 2. Materials and Methods

### 2.1. Ethical and Subject Selection

This prospective, randomized, split-mouth, single blinded trial followed the CONSORT guidelines. Ethical acceptance obtained by Siksha ‘O’ Anusandhan (deemed to be) University Ref.no/DMR/IMS/SOA/180417, Clinical trial registration No CTRI/2020/09/027559 and Patent Application 201831039005.

Sixty subjects aged 18–45 years old (60 occlusal caries teeth and 60 proximal caries teeth) enrolled voluntarily, and the study was conducted at the Institute of Dental Sciences, Siksha ‘O’ Anusandhan (deemed to be) University, in accordance with the principles of the Declaration of Helsinki, 2008. All subjects were told the purpose and design of the investigation, an informed written consent form was signed, and each received routine dental treatment. A detailed medical and dental history was recorded. The commencement of study was from August 2020. Enrollment began in September 2020.

### 2.2. Sample Size

The sample size was determined using G* Power software, version 3.1.9 (available at http://www.gpower.hhu.de/en.html (accessed on 15 January 2020), considering a split mouth design. The selected test details were significant (*p* = 0.05): test power (1-b); drop-out = 0.10. The final sample size was a total of 120 teeth from *n* = 60 patients with occlusal and proximal caries of *n* = 60 teeth each (two teeth in each patient) [7,8,9]. Inclusion criteria: Age: 18–45 years old with no systemic illness; active, acute dentinal caries involving the occlusal surface and/or proximal surface of at least two permanent molars and premolars, not surpassing 2/3rd of the dentin (D1, D2, D3 radiographical baseline); with direct access and view; and no clinical or radiographic signs or symptoms of pulp involvement. Exclusion criteria: patients with systemic diseases; intraoral/extraoral swellings; dental caries involving the pulp; and any periapical pathology.

### 2.3. Study Design

Two weeks of extensive training was conducted, where the operators and evaluators performed the clinical procedures of the study, and became expert at evaluating the cavity with complete and incomplete caries removal. After the screening process, patients were randomly allocated (computer generated software available at www.randomizer.org accessed on 10 September 2020) The numbers were distributed in envelopes by a researcher who did not participate in the clinical stage, and for each subject, one tooth was randomly selected while other teeth automatically received the other treatment. This randomization resulted in 120 teeth from patients with an occlusal (O) and/or proximal (P) carious lesion *n* = 60 each (2 teeth per patient; Figure 1).

Group A (*n* = 60) Occlusal group: Group A1 (control) (*n* = 30), and Group A2 (BCD) (*n* = 30).

Group B (*n* = 60) Proximal group: Group B1 (control) (*n* = 30) and Group B2 (BCD) (*n* = 30).

### 2.4. Treatment

Pre-operative, during, and post restoration clinical photographs, bitewing RVGs (NanoPix1, Intraoral Digital Imaging Sensor, APS CMOS technology, Eighteeth, Caretechion GmbH, Duesseldorf, Germany) were taken, and silicon impression (Extreme Putty, Medicept UK LTD, Middlesex, UK) of pre-excavated lesions was made for calibration of caries extension. A single trained operator initiated the procedure, and local anesthesia was administered if needed. Clinically, two measurements were made using a periodontal probe (Api UNC15). 1. The outer diameter of the carious lesion was measured bucco-lingually and mesio-distally, and 2. the depth of the lesion was measured at three different sites, pulpally (when possible before excavation). After assessing the cavity with a periodontal probe, silicon impressions (Extreme Putty, Medicept UK LTD, Middlesex, UK) of post excavation cavity depth were made to determine cavity volume.

Cavity volume was evaluated by 3D scan analysis (EXOCAD Software Medite scanner) as shown in Figure 2. The treatment time of caries excavation was measured. A patch test for BCD group was performed. The procedures were performed under 3.2× loupe (Admetec Solutions LTD, TTL 3.2× Moris Sweden and Orchid Light Haifa, Israel) and the cavity was finished with micro-grained aluminium oxide abrasive Arkansas stone (RD2 &TC1 (Shofu Dura-White Stones, Kyoto, Japan)) at a speed range 5000–20,000 rpm and contact pressure 2.

#### Caries Excavation and Treatment Time

For both occlusal (A) and proximal groups (B), the carious lesions of the control group (group A1, B1) were excavated using 311RM, 440S, 440, and 462R diamond points (Shofu Inc Diamond Points Fg, Kyoto, Japan) and an air-rotor handpiece (NSK PANA-AIR MB2 CB90036, Japan) at a speed of 100,000 to 150,000 rpm. For Groups A2 and B2, 0.5 mL of BCD solution applied and scrubbed with a micro-brush for 20 s, and a small pellet of damp cotton was placed into the cavity. RVG was taken after 40 s of mechanical excavation performed with a sharp spoon excavator (Mcare Instrument, XmsH, Maharashtra, India). Repeated application of solution was carried out until effective caries removal was attained. Both groups were judged by standard clinical criteria, and pre- and post-excavation images were measured using digital RVG software. Clinical photographs and silicon impressions were made, and complete caries removal was assured by evaluating the color and consistency (probing of the excavated cavity will be undertaken until a hard surface felt) and scored (0–5) according to the Ercison D et al. (1999) scale for caries assessment. Working time was recorded using a digital stop-clock. The digital stop-clock started when the first step of excavation or the starting of cavity preparation and stopped when the caries excavation and cavity preparation was completed. Occlusal restorations were completed with resin-modified glass ionomer cement (RMGIC) (XtraCem-LC, Medicept UK LTD, Middlesex, UK) and light cured for 10 s in the low mode (Bluephase® N IvoclarVivadent, Schaan, Liechtein) and the restored surface was protected with bonding agent (Tetric® N-Bond Universal, IvoclarVivadent, Schaan, Liechtein).

In the case of proximal caries group, a sectional matrix band (TDV Special Matrices Kit, TDV Dental Ltd. Pomerode/SC, Brasil) was used to replace the missing proximal wall, and cavities were restored using bonding agent (Tetric® N-Bond Universal, Ivoclar Vivadent, Schaan, Liechtein), bulk fill Composite (Tetric^®^ N-Ceram Bulk Fill, Ivoclar Vivadent, Schaan, Liechtein), light cured for 20 s in the soft mode (Bluephase® N Ivoclar Vivadent, Schaan, Liechtein) and sandwich technique (in deep cavities approximating pulp) (Ionosit, DMG CPF GmbH, Hamburg, Germany) was performed. Subsequently, for both groups, occlusion correction and finishing and polishing of restorations (Optra gloss® Ivoclar Vivadent, Schaan, Liechtein) (EVE Diacompplus Occluflex, Keltern, Germany) were performed. Post-restoration was evaluated by using a clinical photograph and radiograph (Figure 3, Figure 4 and Figure 5).

### 2.5. Evaluations

Radiographic evaluation: the preoperative radiographic evaluation of caries extension was performed by measuring the remaining dentin thickness after using color and revealer mode in the RVG software. With bitewing digital RVG software (70 kv for 0.40 s exposure time), the solution was evaluated as radiopaque after application. Bitewing or periapical radiographs were used for assessing immediate restoration and at 3, 6, and 12 months post-restorations according to Federation Dentaire Internationale criteria [8]. Post-restoration was assessed for overhanging restoration, recurrent caries, defects/voids in restoration, periapical or periradicular radiolucency, or violation of biological width [10].

Clinical Evaluation: The calibrated (Kappa statistic: 0.92) and trained single operator evaluated the caries excavation under a microscope (10x magnification, Sanma Dental Microscope, Sanma Medineers vision Pvt LTD. Chennai, India), and clinical digital photographs (DSLR Canon 77D). The color scoring interpretation of each tooth was identified according to Y. Hosoya 2007 [11] as: (1) black-colored group (black or dark brown), (2) brown-colored group (brown or yellow brown), and (3) yellow-colored group (yellow or light yellow). Caries hardness classification was (a) hard, (b) medium, and (c) soft. (Hard: carious dentin required the use of both burs and hand instruments for removal; soft: caries-infected dentin was easily removed with hand instruments; medium: carious dentin fell between the aforementioned categories). Overall assessment of caries was scored according to Ericson D et al., 1999 [12] as given below: 1. Caries removed completely; 2. caries present in base of cavity; 3. caries present in base and/or one wall; 4. caries present in base and/or 2-wall; 5. caries present in base and/or more than 2-wall.

Measurement of the volume of carious tissue excavated: post-caries excavation impressions was made by putty single-step elastomeric addition silicone material (Extreme Putty, Medicept UK LTD, Middlesex, UK) in sectional impression tray, according to direction for use (DFU) [13]. The cast was prepared using type IV die stone (Pearl stone, Asian Chemicals, India). Differences between initial and final cavity volumes were determined using 3D scanner MEDIT T500 (MEDIT corp. 23 Goryeodae-ro 22 Gil, Seongbuk-gu, Seoul, South Korea) and Exocad Software (software version; exocad Dental DB 2.3 Matera 6990, Darmstadt, Germany). The three-dimensional morphology of the excavated cavity was obtained using a 3D laser scanning microscope using phase-shifting optical triangulation with an LED light source (150 ANSI-Lumens, blue LED). In the prepared surface field, by selecting a point spacing of 0.04 mm and scanning an area of 100 mm × 73 mm × 60 mm, the supporting software automatically calculated the enclosed volume. (Figure 2)

Pain assessment by Visual Analog Score (VAS), VAS for facial expression, and Sound Eye Motor scale (SEM): VAS Lego pain assessment tool used was created by Brendan Powell Smith with ratings from 0–10: no pain, mild pain, moderate pain, serious pain, severe pain, and worst pain; along with emotional face pictorial scales which included: alert smiling, no humor, serious flat face, frowning sad eyebrows, intense stare grievance, bulged eyes, audible screams, palpable fear, and agonizing screams with faces distorted beyond recognition. A pain reaction using the sound, eye, and motor scale (SEM) was recorded according to Wright et al., 1999 [14]. For each patient the same investigator recorded the pain assessment at baseline, during treatment, and after treatment. For both groups the baseline was recorded before the initiation of the procedure. The administration of local anesthesia (Lignox2%A Warren Indoco Remedies LTD Navi Mumbai, Pawane, India) was carried out on the basis of patient response and the cause of pain. Scores of three scales were subjected to statistical analysis for the assessment of patient acceptance.

Two experienced calibrated evaluators independently evaluated the post-restorations using mirrors, probes, clinical photographs, and radiographs at 3, 6, and 12 month intervals using [15] FDI evaluation criteria. It has 16 categories on a scale of 1 to 5 which include esthetic, functional, and biological properties. Scoring parameters were: 1—clinically excellent/very good, 2—clinically good, 3—clinically sufficient/satisfactory, 4—clinically unsatisfactory, and 5—clinically poor. When a restoration received a score of 4 or 5 it was recorded as a failure. Cases were considered successful if each of the 16 following FDI criteria presented an individual score of 3 or less: A. Esthetic Properties: Surface luster [Item 1] Surface staining [Item 2] Color stability and translucency [Item 3] Anatomic form [Item 4] B. Functional Properties: Fractures and retention [Item 5] Marginal adaptation [Item 6] Wear [Item 7] Contact point/food impaction [Item 8] Radiographic examination [Item 9] Patient’s view [Item 10]. C. Biological Properties: Postoperative (hyper) sensitivity and tooth vitality [Item 11] Recurrence of caries, erosion, abfraction [Item 12] Tooth integrity [Item 13] Periodontal response [Item 14] Adjacent mucosa [Item 15] Oral and general health [Item 16]. A customized evaluation sheet was used for the evaluation of restorations. Differences between assessments were discussed, and consensus was achieved for preliminary differences.

Statistical Analysis: Descriptive and analytical statistics were performed using SPSS (Statistical Package for Social Sciences) Version 24.0 (IBM Corporation, Chicago, IL, USA). Intra/inter-examiner agreement was determined using the Kappa statistic and was considered excellent (K = 0.92). The data are represented as number with proportions and a mean with a standard deviation. The normality of continuous data was analyzed by the Shapiro–Wilk test. As the data followed a normal distribution, parametric tests were used to analyze the data. The independent sample *t*-test was used to check mean differences. The chi-square test for independence was used to determine whether two variables in a contingency table are related (*p* < 0.05).

## 3. Results

In Group A, the mean age was 31.76 ± 6.11 years with 19 (63.3%) males and 11 (36.7%) females. There were 35 (58.3%) distal proximal caries and 25 (41.7%) mesial proximal caries in Group B, and had a mean age of 31.20 ± 5.41 years and 16 (53.3%) males and 14 (46.7%) females.

Unexpected excavation of sound dentin with caries was seen in the control group compared to the BCD group. In occlusal group, caries excavation time was 197.90 ± 30.97 s (3.30 ± 0.51 min), 292.13 ± 44.87 s (4.86 ± 0.74 min) and proximal group excavation time was 273.06 ± 69.95 s (4.5 ± 1.16 min), 411.86 ± 88.34 s (6.86 ± 1.47 min), respectively (Table 1 and Table 2).

Caries volume, caries removal score, color score, and consistency: the mean caries removal volume (occlusal and proximal) for the BCD group (0.82 ± 0.31) (28.79 ± 15.39) was significantly less compared to the control group (1.10 ± 0.39), (32.41 ± 18.72) and there was statistically insignificant distribution of caries removal between the two groups. Clinically, compared to control group, 9 teeth in the proximal B2 group were scored 1 (caries present on the base of the cavity), brown, and hard in consistency. The correlation between time (seconds) and volume for the occlusal group (*p* = 0.354) was statistically insignificant. In the proximal group, it was statistically significant (*p* = 0.011): weak positive correlation R = 0.326 was found between time (seconds) and volume. In the BCD group, during the procedure, the mean VAS, FACE, and SEM (Group A2 (1.96 ± 1.71), (1.76 ± 0.81), (1.90 ± 1.06) and Group B2 (2.73 ± 1.20), (2.56 ± 0.56), (2.03 ± 0.76)) were significantly less than the Control group (Group A1 (4.03 ± 1.77), (2.50 ± 1.04), (2.80 ± 0.96) and group B1 (4.23 ± 1.59), (3.20 ± 0.80), (3.03 ± 0.88), respectively).

Immediately post restoration and after 24 h responses were statistically insignificant. Pain perception or experience was significantly less in BCD (A2, B2) and acceptability was higher than in the control group (A1, B1).

Post restoration evaluation (FDI Criteria) was clinically (aesthetic, functional, and biological) and radiographically excellent in group A at 3 months (*p* = 0.247), (*p* = 0.065), (*p* = 0.600), 30(100.0%); 6 months (*p* = 0.330), (*p* = 0.783), (*p* = 0.726), (*p* = 0.317); and 12 months (*p* = 0.489), (*p* = 0.429), (*p* = 0.091), (*p* = 0.150), respectively (Table 3). In group B, a statistically insignificant difference in the distribution of clinical and radiographical properties at 3 months (*p* = 0.299), (*p* = 0.300) (*p* = 0.232) (*p* = 0.495); 6 months (*p* = 0.594) (*p* = 0.947) (*p* = 0.953) (*p* = 0.274) and 12 months (*p* = 0.494), (*p* = 0.429) (*p* = 0.675) (*p* = 0.740) was found, respectively (Table 4). The overall postoperative radiographic evaluation was significant for BCD as compared to the control group (Table 3 and Table 4). For both occlusal and proximal groups, caries volume was significantly low in the BCD group (Table 1 and Table 2).

The dropout rate during follow-up evaluation was as follows: at 3 months and 6 months, 60 patients with 120 teeth were examined and scored as per FDI criteria. Three patients visited at 4 months and 2 patients within a 7 months interval due to travel inconvenience, but these patients were in virtual consultation at 3 months and 6 months. Five patients were considered for evaluation to prevent post-randomization bias and avoid affecting the power of the study. Six patients’ data were lost at a 12-month interval due to unavoidable circumstances. It was noted that no significant difference was found between the delayed evaluation of the two groups.

## 4. Discussion

The split mouth design for occlusal and proximal caries was considered in the current study to analyze and understand the procedural steps and pain perception for the comparison of the conventional method with the BCD group. This design was chosen because it has uniformity (each individual receives all treatment/intervention modalities); it eliminates the impact of inter-individual variations related to patient demographic characteristics and systemic health conditions, including genetic susceptibility and oral hygiene status; and because there are clinically significant between-site differences related to the outcome of interest, allowing for a powerful estimation of treatment effects [16]. Because this was a new formulation, the operator and evaluator were trained, and the procedure was explained before the trial. The BCD in vitro studies that were used as a reference provided information on the mode of application [6,17]. Patient blinding was not possible because we had to explain the procedure to the patient and conduct an allergy patch test for BCD solution on the inner surface of the arm. None of the patients experienced hypersensitivity. The patient was treated in both quadrants at random from the treatment groups [9]. However, the operator did not have access to the evaluated data because the professional operator was not the same as the evaluator. The operator had no access to patient information, whereas the evaluator had access to all of the patients’ responses and was aware of the treatment chosen. It is much easier to deal with occlusal caries than proximal caries, which necessitate the removal of impacted food, control of gingival bleeding, accessibility, development of contact and contour, overlapping of surfaces on X-ray, nicking of adjacent teeth (control group), prevention of gingival marginal gap, and management of distal proximal surface caries, which is usually challenging.

In the current study, the average age for both groups was 30–35 years, with males having more caries experience than females. Literature suggests that females have a higher caries rate than males [18,19], and a study found that adult men have more new caries than females, implying a gender bias in accessing or utilizing dental health services. Our findings were consistent with those of Shaffer et al., 2015 [20].

In this study, a novel formulation BCD is a radiopaque chemochemical solution that aids in the identification and removal of caries through chemochemical action. BCD is a mixture of Iobitridol, 3% Chitosan, and 15% nanohydorxyappatite, and laccaic acid with a pH between 6.5 and 7. After application, the dye component of BCD aids in clinically identifying caries, as compared to any caries-disclosing dye, which stains the denatured dentine and has the disadvantage of being ineffective in differentiating caries radiographically. However, the BCD component Iobitridol detects the extent of caries radiographically, which aids in treatment decision-making. In the literature, there is little evidence of radiopaque chemomechanical or chemochemical solutions in the commercial market. Due to the superimposition of anatomical structures and artifacts, intra-operative radiographs have low sensitivity and an unacceptable proportion of false-negative results in the detection of dental caries [21]. BCD solution can make the deep extension of a radiolucent lesion appear radiopaque, allowing for a clear assessment of lesion margins. As there was limited extension of BCD on RVG in the proximal group, complete removal of food particles was performed before placing the BCD solution. Comparative studies with Carie-Care revealed that BCD had a significantly higher zone of inhibition and synergetic effect in the clinical and radiographic identification of dental caries [17]. Various authors have discussed the positive and false positive outcomes of CMCR (Carisov, Carie care, Papacarie) [5,22,23].

The total time taken in the BCD occlusal group ranged between 4 and 5 min compared to the control group of 3–4 min, and in the proximal group of 6–7 min and 4–5 min, respectively. The diamond points and carbide burs have high cutting efficiency, and they create less heat generation compared to stainless steel burs [24]. Evidence has proved that the conventional rotary method (diamond points or carbide burs) is preferred by clinicians in the excavation of caries as it consumes less working time but has the drawback of over preparation and removal of reversibly affected dentin, odontobalastic reaction zone plugs, and sound tooth structure, thus irreversibly resulting in the exposure of permeable healthy dentin [25]. According to a study, the hardness of the sound, carious dentine, and arrested carious dentin is 54–65 KHN, 20 KHN or less, and 39.2 KHN, respectively, and after excavation with a spoon excavator, the average hardness of the remaining dentin is 23 KHN, leaving soft dentin thickness ≥0.7 millimeters [26]. Previous caries excavation studies shown that time consumed by various CMCR is relatively high [25,27] when compared to BCD solution.

The number of applications of BCD depended on the consistency of the carious enamel and dentin. In the Occlusal caries group, a single application of BCD for 20–40 s could remove around 60–80% of caries. Sometimes near DEJ and the base of the cavity, two applications were required for excavation. Hard caries or arrested caries were difficult to excavate, and BCD did not stain sound dentin or arrested dentin rather, it helped in distinguishing between soft dentin and sound tooth structure. In the proximal group more than two applications were required to remove caries from the inner wall of the cavities. Small size spoon excavator was used in D2 caries extension, and the irregular cavo-surface was smoothed by Arkansas stone. The stone removes unsupported enamel and finishes without removal of excess sound enamel margins. In the control group B1, to avoid nicking of adjacent tooth, a cut matrix band was used during cavity preparation. This step was eliminated in BCD group B2. In some teeth, wedges and retraction cords were used interdentally during caries excavation to prevent gingival bleeding and retraction of gingiva to expose the gingival margins. Under magnification (Loupes/DOM) with minimal cutting caries were excavated for both groups effectively. Magnification helped in control-selective removal of caries [28].

It was observed that 34% of V type occlusal fissure, wide at the top and gradually narrowing towards the bottom of the fissure and 26% of IK type with a narrow space associated with a larger space at the bottom, were of concern [29,30]. Easy caries excavation was observed in open dentinal caries compared with layered or covered with enamel. In the proximal lesions, absence of marginal ridges or open cavities was easy to approach rather than the presence of intact undermined enamel marginal ridge. Ideal caries excavation technique removes the irreversibly denatured tissue, leaving behind the potential remineralizable tissue at the floor of the cavity. Complete caries removal was judged on the basis that the explorers should unstick in the dentin, with no tug-back sensation, brown-hard surface, and surface should be stain free [31]. The evaluator assessed the case and if the case did not meet the criteria, the procedure was repeated. The results of this study showed that effective caries removal was seen in both groups. In the control group the abrading action of diamond points and speed of the air-rotor helped in caries excavation, while for the BCD group, application on the carious surface with micro-brush and scrubbing for 40 s helped in softening the degraded collagen and provided guided and gentle excavation with a spoon excavator. Active ingredient 3% CS (molecular weight: 161.16 g/mol) is a polycationic, good chelating agent with a strong ability to bond with proteins and a regenerative effect on connective tissues. Nanohydroxyapatite has the potential to remineralize the enamel and dentine caries lesions. The combinatorial effect has been explained in our previous studies [3,6,32].

In the current study, the procedure of impression making of the prepared/excavated cavity is in reference to the study [13]. A 3D scanner determined the volume of caries removed for both groups. The Control groups A and B showed high volume compared to the BCD group, thus stating that more sound tooth structure was removed in the Control group while there was preservation and a chance to remineralize the excavated surface in case of the BCD group. The scanning was performed directly from the impression and cast. It was observed that there was a negligible difference in mm between impression and cast scanning. Direct 3D scanning of the impression takes less time and provides more accurate volume determination than scanning the cast (extra time to fabricate the cast, contraction/expansion of the dental stone material, and porosity inclusion) [33]. Micro-CT studies revealed that the conventional rotary method produced a large volume of post caries excavation cavity compared to CMCR methods [34,35].

In reference to the in vitro study, BCD containing chitosan and nHA together create a homogeneous compound with high strength and adhesion properties [36], and their dual combination in BCD imparts a protective effect against secondary caries and has corresponding antibacterial properties with influence on the reduction of DH [6]. Previous studies suggested that the conventional rotary procedure produces sound, vibrations, and heat generation affecting patient mindset and behavior [24,37]. No patient in the BCD group experienced pain or received anesthesia. However, patients in the Control group with four occlusal caries teeth and seven proximal caries teeth insisted on LA during excavation. There was a significant positive correlation (*p* = 0.001) found between the VAS, FACE, and SEM for the occlusal group and a strong positive correlation (*p* < 0.001) for the proximal group during treatment. There was no statistically significant difference found after the treatment between the groups.

In Group A (occlusal caries), RMGIC was used as restorative material. RM-GIC (1990s) adhesive contains PAA (polyacrylic acid), has dual curing properties, and can completely set in total darkness within 24 h. It has advantages of micro-mechanical and chemical bonding to the tooth structure by forming ionic bonds between carboxyl groups of RMGICs and calcium ions of enamel and dentin [15,38]. This material is superior to conventional GIC; hence, it was considered for restoration in the occlusal group.

In Group B (proximal caries), bulk fill composite restoration was considered for two reasons. Firstly, RMGIC has a major drawback deterioration in color match/stability, surface roughness, anatomic form, compromised marginal adaptation, and marginal ridge strength as to withstanding the masticatory forces, while composite are superior aesthetically and in mechanical properties [10,39,40,41]. The second reason is to evaluate the survival rate of the composite restorations after the use of BCD.

According to FDI criteria, Aesthetic: The results of the occlusal group A2 showed that 53.3%, 6.7% of restorations were ‘clinically good’; 0.0%, 6.7% ‘clinically unsatisfactory’; and 3.3%, 20.0% ‘clinically poor’, and in the Control group A1, 36.7%, 20.0% were ‘clinically good’; 13.3%, 0.0% ‘clinically unsatisfactory’; and 6.7%, 23.3% ‘clinically poor’ at 3, 6, and 12 months, respectively. At 3 months, the Control group had 16.7% fewer cases in the ‘clinically excellent’ parameter, while BCD had 20.0%; at 6 months, the control group had 46.7% and BCD had 50.0%. It was observed that parameter 1 significantly increased for both groups over a period. The probable reason was rectifying or repairing as per the items 1 (Surface luster) 2 (Surface staining), 3 (Color stability and translucency) improves the survival rate and increases the patient awareness. The obtained results signify that BCD solution does not affect the survival rate of the RMGIC restoration (Table 3).

For the proximal group, the Control group B1 showed 63.3%, 66.7% ‘clinically excellent’; 26.7%, 10.0% ‘clinically good’; 3.3% ‘clinically unsatisfactory’; 6.7% ‘clinically poor’; and BCD B2 showed 40.0%, 63.30%; 46.7%, 3.30%; 6.7%; 3.3%, and 16.70% at 3, 6, and 12 months intervals, respectively (Table 4).

Three teeth in both groups showed rough, small voids at the tooth restoration margin, but at 6 and 12 months intervals 10 teeth (Control: 4 and BCD: 6) needed repair or re-restoration. The results signify that the BCD solution does not affect the survival rate of the composite restoration.

Functional: Occlusal caries group, in the Control group, 40.0%, 66.7% were ‘clinically excellent’; 53.3%, 10.0% ‘clinically good’; 6.6%, 0.0% ‘clinically satisfactory’; 0.0%, 10.0% ‘clinically unsatisfactory’; 0.0%, 13.3% ‘clinically poor’; and in BCD group, 70.0%, 56.7%; 26.7%, 16.7%; 3.3%; 0.0%, 13.3% at 3 and 6 months intervals, respectively. Radiographically, results were 100% excellent for both groups at 3 months, and control group 13.3% and BCD 23.3% at 6 months. Initial disparity in restorations was not well appreciated in the radiographs. Both groups performed functionally well without any fracture of RMGIC (Table 3).

In the proximal caries group, control group, 73.3%, 53.3% ‘clinically excellent’; 13.3%, 16.7%, ’clinically good’; 3.3%, 10.0% ‘clinically unsatisfactory’; 10%, 16.7% ‘clinically poor’, and in the BCD group, 83.3%, 53.3%; 10%, 10.0%; 6.6%, 13.3%; 0.0%, and 20.0% at 3 month and 6 months intervals, respectively. Radiographically, in control group, 89.7%, 73.3% ‘clinically excellent’; 6.9% ‘clinically unsatisfactory’; 6.6%, 20% ‘clinically poor’, and BCD group 93.3%, 63.3%; 0.0%, 0.0%; 3.3%, and 20%, respectively (Table 4).

Items 5 (Fractures and retention), 6 (Marginal adaptation), 7 (Wear), 8 (Contact point/food impact) were noticed frequently. In the BCD group, 10 teeth showed partial/complete loss of restoration compared to the Control group’s 8 teeth at a 6 months interval. Marginal ridge fracture of composite restoration and food lodgment was seen in most cases. Apart from internal associated factors during the procedure, further investigation is needed to find the compatibility of the BCD solution with the bonding agent. Re-restorations were performed for both groups.

Biological: In the occlusal caries group, control group, 73.3%, 63.3% ‘clinically excellent’; 23.3%, 10% ‘clinically good’; 3.3%, 3.3% ‘clinically satisfactory’; 0.0%, 13.3% ‘clinically unsatisfactory’; 0.0%, 10.0% ‘clinically poor’, and in the BCD group, 76.7%, 56.7%; 23.3%, 10.0%; 0.0%, 13.3%; 0.0%, 10.0%; and 0.0%, 10.0%, respectively. The patient did not experience post-operative sensitivity for both groups. BCD solution has the potential to enhance the bonding interface of the tooth and RMGIC, as it consists of nanohydroxyapatite, one of the components of the tooth structure [17] (Table 3).

The proximal caries group, in the control group 83.30%, 60.0% ‘clinically excellent’; 6.70%, 13.3% ‘clinically good’; 0.0%, 6.70% ‘clinically satisfactory’; 6.7%, 13.30% ‘clinically unsatisfactory’; 3.3%, 6.70% ‘clinically poor’ and in BCD group, 83.30%, 60.0%; 16.70%, 10.0%; 0.0%, 10.0%; 0.0%, 10.0%; and 0.0%, 10.0% at 3 months and 6-month interval, respectively. Item 11 (postoperative (hyper) sensitivity and tooth vitality), item 12 (periodontal response), 6 teeth from both groups developed sensitivity at 6 months interval due to the partial/complete loss of the composite restoration. Sensitivity subsided after restoring with composite. The results were clinically excellent and corroborate both groups (Control and BCD). As a result, the success rate in BCD is more appreciable than the conventional rotary (Table 4).

BCD excavated dentine has good bonding receptivity with RMGC and composite materials, implying that it has the potential to occlude the dentinal tubules by nanohydroxyapatite, reducing the hypersensitivity and creating a bonding surface for restorative materials [17]. Total excavation time of the proximal lesion was longer as compared to the occlusal carious lesion for the BCD group.

The results of the study reject the null hypothesis, and limitations of the study include examining the efficacy of a combination of the two techniques (rotary and BCD) and comparison with commercially available chemochemical solutions. Further clinical trials can be conducted for different parameters to provide strong evidence for the BCD solution as a radiopaque chemochemical solution.

## 5. Conclusions

Overall, the occlusal rotary group had less working time than the BCD group. The BCD group had an increase in working time. The BCD group showed greater patient acceptance and decreased pain during excavation. There was no significant difference between the two groups in terms of color score, surface hardness, and caries removal score. The post-restoration evaluation at 3, 6, and 12 months was acceptable for both groups. Based on these findings, we can conclude that BCD has the potential to identify the caries clinically and radiographically and helps in effective caries removal. It is a potential strategy for treating children, adults, special needs, community-level, and geriatric populations.

## 6. Patents

Intellectual property of India: patent application 201831039005. “Bioactive Dental Detective Dye”.

## Figures and Tables

**Figure 1 biomedicines-11-00500-f001:**
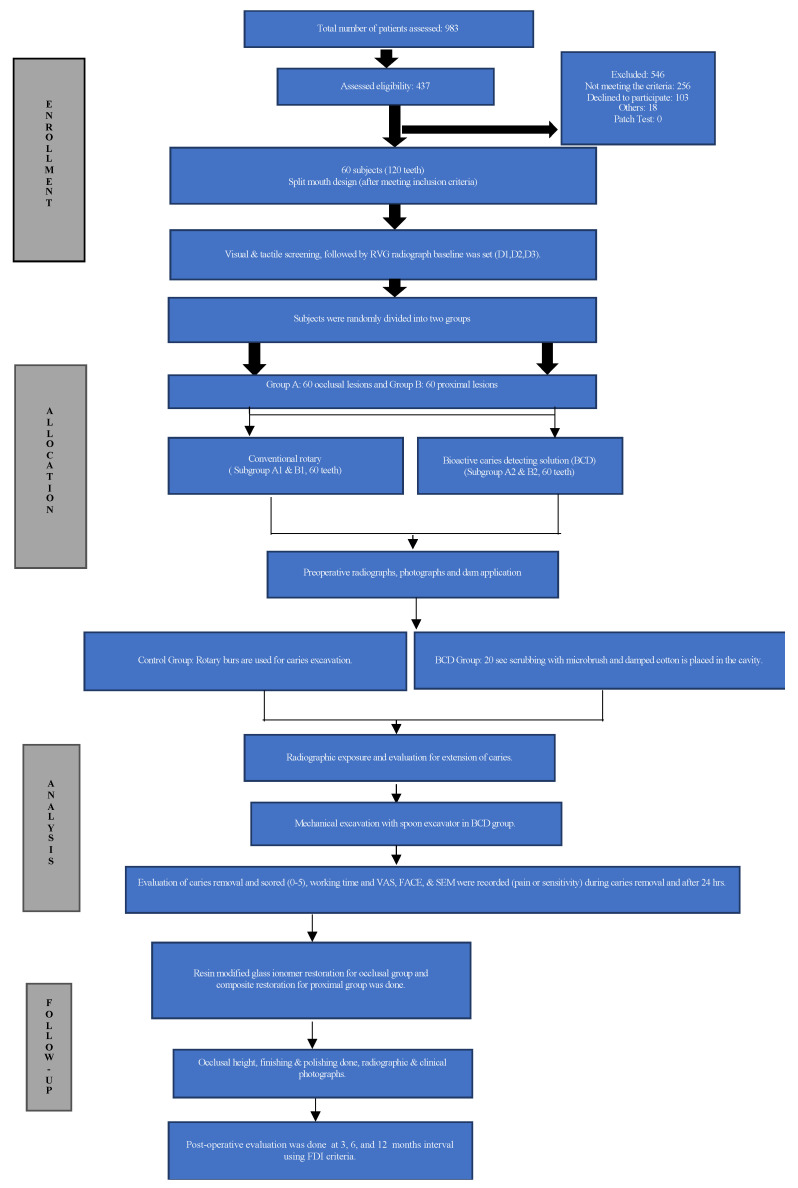
Flow chart of clinical study (CONSORT) for BCD.

**Figure 2 biomedicines-11-00500-f002:**
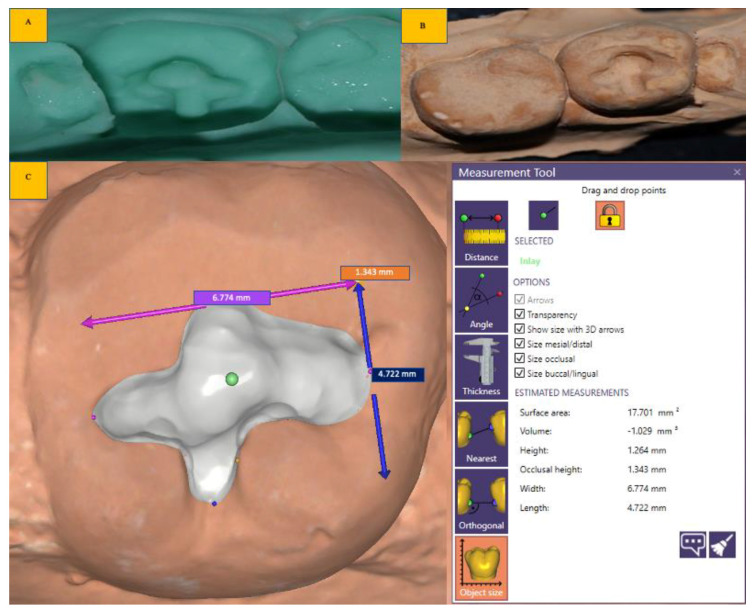
The steps of volume determination by 3D scanner: (**A**): silicon impression; (**B**): working cast; (**C**): volume determination illustration (mm^2^) using exocad software.

**Figure 3 biomedicines-11-00500-f003:**
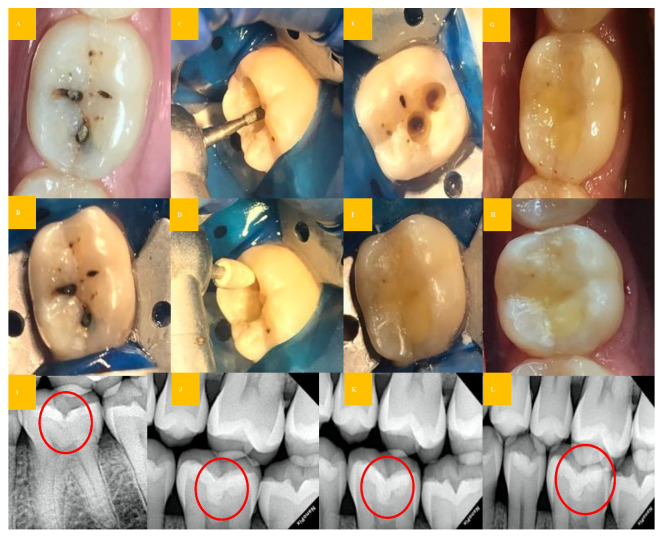
Clinical steps for the occlusal caries group (Control group): (**A**,**B**,**I**): preoperative clinical photo and radiograph (red circle); (**C**): caries excavation with diamond point; (**D**): finishing of cavo-surface margins using Arkansas stone; (**E**): final prepared cavity; (**F**,**J**): cavity restored with RMGIC material and immediate post-restoration radiograph (red circle); (**G**,**K**): post-restoration clinical photo and radiograph (red circle) after 3 months; (**H**,**L**): post-restoration clinical photo and radiograph (red circle) after 6 and 12 months.

**Figure 4 biomedicines-11-00500-f004:**
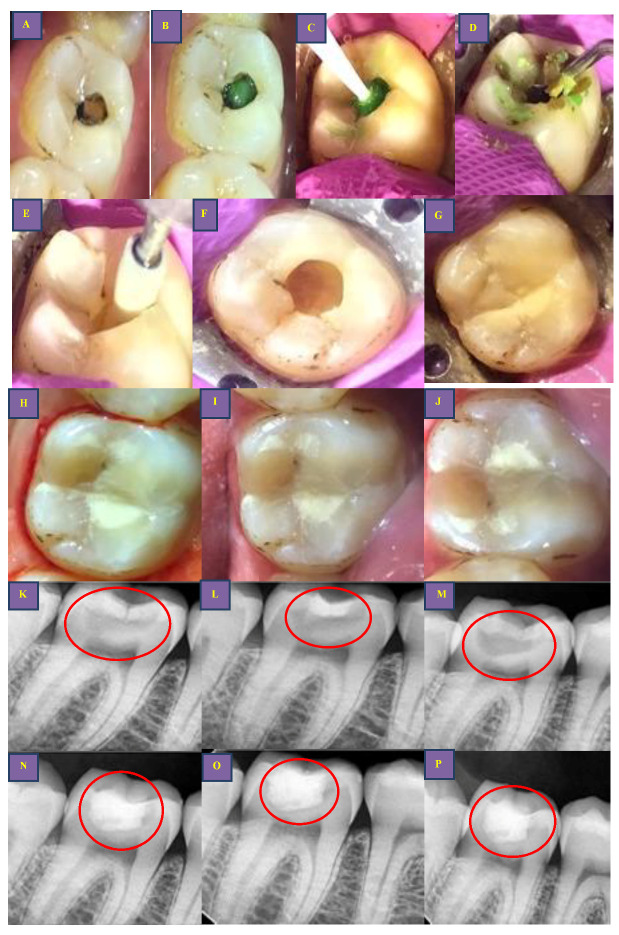
Clinical steps for occlusal caries group (BCD group): (**A**,**K**): preoperative clinical photo and radiograph (red circle); (**B**,**C**,**L**): application clinically and extension of radiopaque BCD on radiograph (red circle); (**D**): caries excavation with spoon excavator and caries debris during excavation; (**E**): Finishing of cavo-surface margins using Arkansas stone; (**F**,**M**): final prepared cavity and post-caries excavation radiograph (red circle); (**G**): cavity restored with RMGIC material: (**H**,**N**): post-restoration clinical photo and radiograph (red circle) after 3 months; (**I**,**O**): post-restoration clinical photo and radiograph (red circle) after 6 months; (**J**,**P**): post-restoration clinical photo and radiograph (red circle) after 12 months.

**Figure 5 biomedicines-11-00500-f005:**
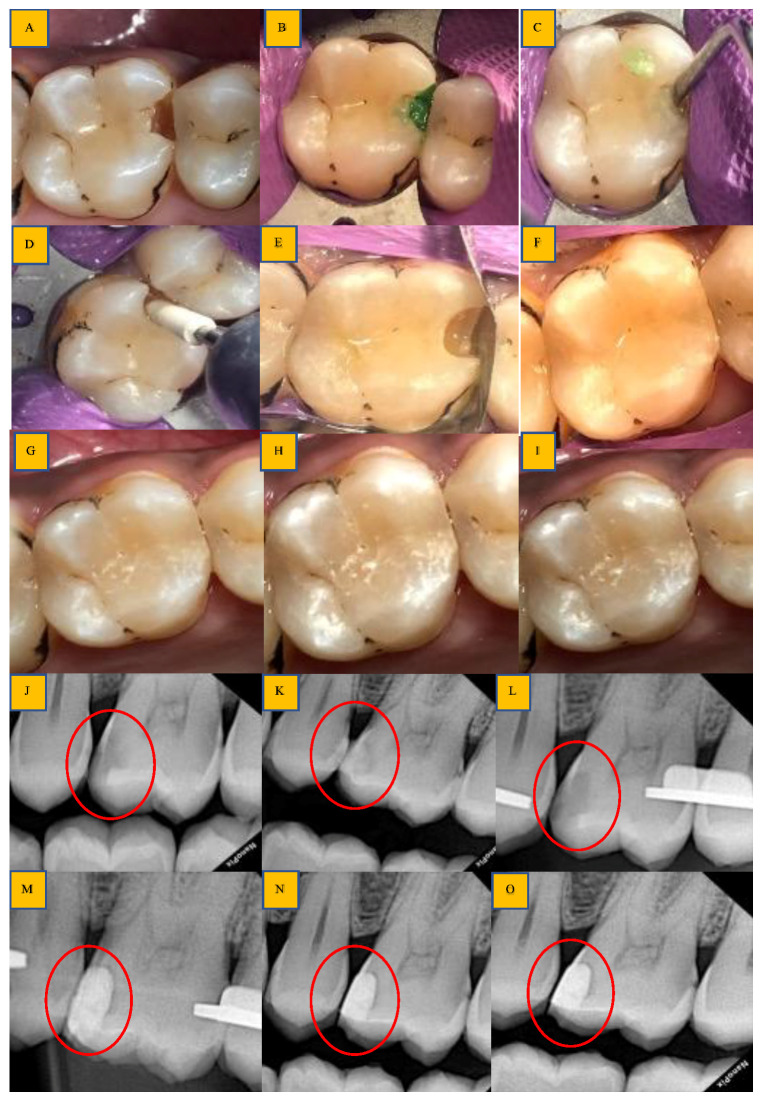
Clinical steps for the proximal caries group (BCD group): (**A**,**J**): preoperative clinical photo and radiograph (red circle); (**B**,**C**): application of BCD solution; (**K**): extension of radiopaque BCD solution seen on radiograph (red circle); (**D**,**E**): caries excavation with spoon excavator and finishing with Arkansas stone; (**F**,**L**): post-caries excavation clinically and radiographically (red circle); (**G**,**M**): cavity restored with composite material and immediate post-restoration radiograph (red circle); (**H**,**N**): post-restoration clinical photo and radiograph (red circle) after 3 months; (**I**,**O**): post-restoration clinical photo and radiograph (red circle) after 6 and 12 months.

**Table 1 biomedicines-11-00500-t001:** Mean time, caries removal score, color score, consistency score, volume, mean VAS, FACE, and SEM at baseline, during procedure, after completion and at 24 h among the groups (Occlusal caries).

Occlusal	Control		BCD	*p*-Value
**Mean time (S)**	197.90		292.13	<0.001 ^†^
**Caries removal score**	Score 0	24 (80.0)	22 (73.3)	0.542
**Color score**	Black Brown Yellow	2 (6.7) 4 (13.3) 24 (80.0)	3 (10.0) 5 (16.7) 22 (73.3)	0.820
**Consistency**	Hard	30 (100.0)	30 (100.0)	-
**Mean carious removal volume (mm^3^)**	1.10		0.82	0.003 ^†^
**Mean VAS score**	During	4.03	1.96	<0.001 ^†^
**Mean FACE score**	During	2.50	1.76	0.004
**Mean SEM score**	During	2.80	1.90	<0.001 ^†^

*p*-value derived from independent sample *t*-test; ^†^ significant at *p* < 0.05.

**Table 2 biomedicines-11-00500-t002:** Mean time, caries removal score, color score, consistency score and volume, mean VAS, FACE, and SEM at baseline, during procedure, after completion and at 24 h among the groups (Proximal caries).

Proximal		Control	BCD	*p*-Value
**Mean time (S)**		273.06	411.86	<0.001 ^†^
**Caries removal score**	Score 0 Score 1 Score 2	23 (76.7) 5 (16.7) 2 (6.7)	20 (66.7) 9 (30.0) 1 (3.3)	0.432
**Color score**	Black Brown Yellow	0 (0.0) 7 (23.3) 23 (76.7)	0 (0.0) 11 (36.7) 19 (63.3)	0.260
**Consistency**	Hard	30 (100.0)	30 (100.0)	-
**Mean carious removal volume (mm^3^)**		32.41	28.79	0.417
**Mean VAS score**	During	4.23	2.73	<0.001 ^†^
**Mean FACE score**	During	3.20	2.56	0.001 ^†^
**Mean SEM score**	During	3.03	2.03	<0.001 ^†^

*p*-value derived from independent sample *t*-test; ^†^ significant at *p* < 0.05.

**Table 3 biomedicines-11-00500-t003:** Aesthetic, functional, biological, and radiological post-operative evaluation at 3, 6, and 12 months among the groups (Occlusal caries).

Occlusal	Period (Months)	Clinically Excellent		Clinically Good		Clinically Satisfactory		Clinically Unsatisfactory		Clinically Poor		*p*-Value
		Control	BCD	Control	BCD	Control	BCD	Control	BCD	Control	BCD	
**Aesthetic**	3	5 (16.7)	6 (20.0)	11 (36.7)	16 (53.3)	8 (26.7)	7 (23.3)	4 (13.3)	0 (0.0)	2 (6.7)	1 (3.3)	0.247
	6	14 (46.7)	15 (50.0)	6 (20.0)	2 (6.7)	3 (10.0)	5 (16.7)	0 (0.0)	2 (6.7)	7 (23.3)	6 (20.0)	0.330
	12	10 (37.0)	13 (48.1)	15 (55.6)	14 (51.9)	0 (0.0)	0 (0.0)	1 (3.7)	0 (0.0)	1 (3.7)	0 (0.0)	0.489
**Functional**	3	12 (40.0)	21 (70.0)	16 (53.3)	8 (26.7)	2 (6.7)	1 (3.3)	0 (0.0)	0 (0.0)	0 (0.0)	0 (0.0)	0.065
	6	20 (66.7)	17 (56.7)	3 (10.0)	5 (16.7)	0 (0.0)	1 (3.3)	3 (10.0)	3 (3.3)	4 (13.3)	4 (13.3)	0.783
	12	11 (40.7)	15 (55.6)	14 (51.9)	12 (44.4)	0 (0.0)	0 (0.0)	1 (3.7)	0 (0.0)	1 (3.7)	0 (0.0)	0.429
**Biological**	3	22 (73.3)	23 (76.7)	7 (23.3)	7 (23.3)	1 (3.3)	0 (0.0)	0 (0.0)	0 (0.0)	0 (0.0)	0 (0.0)	0.600
	6	19 (63.3)	17 (56.7)	3 (10.0)	3 (10.0)	1 (3.3)	4 (13.3)	4 (13.3)	3 (10.0)	3 (10.0)	3 (10.0)	0.726
	12	13 (48.1)	20 (74.1)	12 (44.4)	7 (25.9)	0 (0.0)	0 (0.0)	2 (7.4)	0 (0.0)	0 (0.0)	0 (0.0)	0.091
**Radiology**	3	30 (100.0)	30 (100.0)	0 (0.0)	0 (0.0)	0 (0.0)	0 (0.0)	0 (0.0)	0 (0.0)	0 (0.0)	0 (0.0)	-
	6	26 (86.7)	23 (76.7)	0 (0.0)	0 (0.0)	0 (0.0)	0 (0.0)	0 (0.0)	0 (0.0)	4 (13.3)	7 (23.3)	0.317
	12	25 (92.6)	27 (100.0)	0 (0.0)	0 (0.0)	0 (0.0)	0 (0.0)	0 (0.0)	0 (0.0)	2 (7.4)	0 (0.0)	0.150

**Table 4 biomedicines-11-00500-t004:** Aesthetic, functional, biological, and radiological post-operative evaluation at 3, 6, and 12 months among the groups (Proximal caries).

Proximal	Period (months)	Clinically Excellent		Clinically Good		Clinically Satisfactory		Clinically Unsatisfactory		Clinically Poor		*p*-Value
		Control	BCD	Control	BCD	Control	BCD	Control	BCD	Control	BCD	
**Aesthetic**	3	19 (63.3)	12 (40.0)	8 (26.7)	14 (46.7)	0 (0.0)	1 (3.3)	1 (3.3)	2 (6.7)	2 (6.7)	1 (3.3)	0.299
	6	20 (66.7)	19 (63.3)	3 (10.0)	1 (3.3)	3 (10.0)	4 (13.3)	2 (6.7)	1 (3.3)	2 (6.7)	5 (16.7)	0.594
	12	14 (51.9)	13 (48.1)	11 (40.7)	14 (51.9)	1 (3.7)	0 (0.0)	0 (0.0)	0 (0.0)	1 (3.7)	0 (0.0)	0.494
**Functional**	3	22 (73.3)	25 (83.3)	4 (13.3)	3 (10.0)	0 (0.0)	0 (0.0)	1 (3.3)	2 (6.7)	3 (10.0)	0 (0.0)	0.300
	6	16 (53.3)	16 (53.3)	5 (16.7)	3 (10.0)	1 (3.3)	1 (3.3)	3 (10.0)	4 (13.3)	5 (16.7)	6 (20.0)	0.947
	12	13 (48.1)	11 (40.7)	11 (40.7)	15 (55.6)	2 (7.4)	1 (3.7)	0 (0.0)	0 (0.0)	1 (3.7)	0 (0.0)	0.549
**Biological**	3	25 (83.3)	25 (83.3)	2 (6.7)	5 (16.7)	0 (0.0)	0 (0.0)	2 (6.7)	0 (0.0)	1 (3.3)	0 (0.0)	0.232
	6	18 (60.0)	18 (60.0)	4 (13.3)	3 (10.0)	2 (6.7)	3 (10.0)	4 (13.3)	3 (10.0)	2 (6.7)	3 (10.0)	0.953
	12	13 (48.1)	11 (40.7)	11 (40.7)	13 (48.1)	2 (7.4)	3 (11.1)	0 (0.0)	0 (0.0)	1 (3.7)	0 (0.0)	0.675
**Radiology**	3	26 (89.7)	28 (93.3)	0 (0.0)	0 (0.0)	0 (0.0)	1 (3.3)	2 (6.9)	0 (0.0)	2 (6.9)	1 (3.3)	0.495
	6	22 (73.3)	19 (63.3)	1 (3.3)	0 (0.0)	1 (3.3)	5 (16.7)	0 (0.0)	0 (0.0)	6 (20.0)	6 (20.0)	0.274
	12	16 (59.3)	15 (55.6)	8 (29.6)	10 (37.0)	2 (7.4)	2 (7.4)	0 (0.0)	0 (0.0)	1 (3.7)	0 (0.0)	0.740

## Data Availability

Requested data from corresponding author.

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
