# Peer review of "Diagnosis and Assessment of Dental Caries Using Novel Bioactive Caries Detecting Dye Solution"

_biomedicines, 2023, doi:10.3390/biomedicines11020500_

Round 1

Reviewer 1 Report

The manuscript is well written, the study was nicely conducted, providing interesting data to the readers.

Only one remark: the quality of photos needs to be improved.

Author Response

Dear Reviewer,

Thank you for your valuable time for reviewing the manuscript.

1.Only one remark: the quality of photos needs to be improved.

We have attached PPT, which includes all the quality pictures. 

Reviewer 2 Report

Manuscript of considerable interest for the dental sector, before evaluating the possible publication it needs a major revision

Abstract: highlight the results more

Few keywords: add some

Introduction: add all the risk factors that induce the increase in the incidence of caries already published on children mdpi since.

Materials and methods well described

Very confusing results: tables not well structured, highlighting statistically significant data

Discussion: add as future objectives the use of biomimetic hydroxyapatite to reduce the incidence of secondary caries, always studied by the research group of the

Prof Scribante published on polymers mdpi

conclusions: rewrite them according to the requested changes

bibliography: add required references

Author Response

Dear Reviewer,

Thank you for your valuable time in reviewing our manuscript.

1.Abstract: highlight the results moreInclusion of results data and highlighted in red in the manuscript.

2.Few keywords: add some.

Added and highlighted.

3.Introduction: add all the risk factors that induce the increase in the incidence of caries already published on children mdpi since.

Necessary citations are included with the data and highlighted.

4. Very confusing results: tables not well structured, highlighting statistically significant data

highlighted as per the suggestion.

5. Discussion: add as future objectives the use of biomimetic hydroxyapatite to reduce the incidence of secondary caries, always studied by the research group of the

Prof Scribante published on polymers mdpi

Included the respective citation in the discussion.

6.conclusions: rewrite them according to the requested changes

modified the conclusion and highlighted

7. bibliography: add required references.

added

Reviewer 3 Report

The proposed research paper concerns modern methods of caries treatment. This is a topic of great utilitarian importance. I see the need for corrections.

·         The introduction is to be extended in order to better motivate research. Please describe in more detail the current state of knowledge regarding the use of CMCR and BCD. In my opinion, creating an introduction based on 3 citations  is definitely insufficient.

·         Points from a to d mentioned as hypothesis are not hypothesis.

·         A null hypothesis is too general – when it is formulated in this way is inherently false and requires no research.

·         Statistical analysis materials must be described in the methodological section: software, methods used.

·         Used BCD – please give more details about t in methodological section. Chemical composition, method of application etc. Now we have only some date in discussion.

·         The results are well presented, interesting and important, however, table 4 needs to be edited: in this form it is hardly legible.

·         The discussion is well written.

·         In my opinion, more detailed conclusions should be formulated.

My comments are minor corrections, from the research side, I believe that the work is interesting and valuable.

Author Response

Dear Reviewer,

Thank you for your valuable time in reviewing the manuscript.

1.The introduction is to be extended in order to better motivate research. Please describe in more detail the current state of knowledge regarding the use of CMCR and BCD. In my opinion, creating an introduction based on 3 citations  is definitely insufficient.

Necessary inclusions are made and highlighted in the manuscript.

2.Points from a to d mentioned as hypothesis are not hypothesis.

Modified and highlighted.

3.A null hypothesis is too general – when it is formulated in this way is inherently false and requires no research.

Necessary changes made and highlighted.

4.Statistical analysis materials must be described in the methodological section: software, methods used.

Included in methodological section.

5.Used BCD – please give more details about t in methodological section. Chemical composition, method of application etc. Now we have only some date in discussion.

Mentioned and highlighted in the methodological section. 

6.The results are well presented, interesting and important, however, table 4 needs to be edited: in this form it is hardly legible.

Modified and made necessary changes.

7. In my opinion, more detailed conclusions should be formulated.

Rewritten the conclusion section.

Round 2

Reviewer 2 Report

The manuscript has been properly reviewed and can be published